# Ultrafast shock synthesis of nanocarbon from a liquid precursor

Michael R. Armstrong[1]*, Rebecca K. Lindsey [1]*, Nir Goldman[1], Michael H. Nielsen [1], Elissaios Stavrou[1], Laurence E. Fried [1], Joseph M. Zaug[1] & Sorin Bastea [1]*

Carbon nanoallotropes are important nanomaterials with unusual properties and promising applications. High pressure synthesis has the potential to open new avenues for controlling and designing their physical and chemical characteristics for a broad range of uses but it remains little understood due to persistent conceptual and experimental challenges, in addition to fundamental physics and chemistry questions that are still unresolved after many decades. Here we demonstrate sub-nanosecond nanocarbon synthesis through the application of laser-induced shock-waves to a prototypical organic carbon-rich liquid precursor— liquid carbon monoxide. Overlapping large-scale molecular dynamics simulations capture the atomistic details of the nanoparticles' formation and evolution in a reactive environment and identify classical evaporation-condensation as the mechanism governing their growth on these time scales.

[1] Lawrence Livermore National Laboratory, 7000 East Avenue, Livermore, CA 94550, USA. *email: armstrong30@llnl.gov; lindsey11@llnl.gov; bastea2@llnl.gov

Carbon nanoallotropes are valuable nanomaterials with myriad industrial and biomedical applications due to their remarkable physical, chemical and mechanical properties, tunable surface properties, high thermal and chemical stability, and low toxicity[1–6]. A wide range of synthesis routes—including combustion, pyrolysis, arc and plasma discharge, vapor deposition, detonation, etc., and a variety of carbon-rich precursors are currently employed or investigated for their production, but many of the fundamental processes that govern the formation, evolution and, characteristics of the final product remain to be fully elucidated. This is especially true for high pressure techniques, which can provide access to efficient synthesis pathways for new carbon nanomaterials but can pose unique experimental and conceptual challenges. Perhaps the most prevalent example is the detonation synthesis of nanodiamond[3,7,8], in which a carbon-rich organic explosive is both the chemical precursor for nanocarbon production and the generator of the strong shock-waves necessary for its synthesis. It is a widely used production method, yet it remains a largely empirical procedure despite decades of physics, chemistry, and materials science research[3,4,7–12]. The underlying nanocarbon condensation process has been postulated to also be effective in shock-compressed carbon-rich materials[10,11].

Dynamic compression has traditionally been employed to determine equations of state for materials under high pressures and temperatures[13–15] in experiments where such conditions are maintained for a sufficiently long time to reach a fully equilibrated final state. Equilibration can occur over picosecond time scales[16,17], but may take substantially longer in systems where phase transitions[18] or material strength[19] play an important role. Shock-induced chemistry can also occur over picosecond scales[13,20,21], but organic materials with an excess of carbon (e.g., many common explosives) are known to exhibit orders of magnitude longer chemistry completion times[9,22]. Consequently, the appearance of carbon precipitates in a carbon-rich material following a tens of GPa shock is often presumed to be an intrinsically slow, ≈100 ns process; this coincides with current experimental observations of detonation events[23,24]. The intrinsic coupling of shock and chemistry in a detonation wave is both a complication and a strong constraint on the fundamental study and practical use of shock-induced carbon condensation (precipitation) in a chemically reactive system.

Herein we demonstrate sub-nanosecond nanocarbon synthesis through the application of laser-induced shock-waves[13] to an organic liquid precursor—cryogenic liquid carbon monoxide. We employ in situ optical interferometry to determine the thermodynamic state of the sample under transient compression, and in situ absorption spectroscopy to characterize the formation time scale and volume fraction of carbon nanoparticles. We find the time scale for nanoparticle formation to be no greater than ≈50 ps, and the volume fraction to be ≈10%. Subsequent to the experiment, we study the compression-generated carbon nanoparticles in the experimental cell using Raman spectroscopy, recover nanoparticle samples and characterize them using electron microscopy methods. The recovered nanomaterial shares some of the features of detonation nanocarbon, particularly in terms of particle size. By decoupling the shock generation from the properties of the precursor (an important potential benefit for nanoparticle engineering) and focusing on a prototypical system, we circumvent some of the difficulties of previous studies, and at the same time generate broadly applicable insights that can serve as a major stepping stone for the investigation of more complex materials. The results prove that nanocarbon synthesis can be orders of magnitude faster than it is commonly assumed to be in reacting energetic materials. This rapid synthesis allows us to perform direct large-scale molecular dynamics simulations that reach the thermodynamic conditions and time-scales of the experiments and capture the atomistic details of carbon nanoparticles formation in a high pressure–high temperature reactive environment, including the early chemistry and the role of the classical evaporation-condensation mechanism in determining nanoparticle growth. Taken together the experimental and simulation results provide a coherent physical picture of nanocarbon condensation, which may serve as a template and guide for practical applications and future studies of this important process.

## Results

**Ultrafast dynamic compression experiment.** A schematic of our ultrafast dynamic compression experimental set-up is shown in Fig. 1. A close-up of the experimental scheme and an example of velocimetry data is shown in Supplementary Fig. 1. As described in detail in the "Methods" section and Armstrong et al.[13,25,26], a ≈1 ns duration broadband, chirped, shaped laser pulse impinges on an aluminum ablator placed on the inside surface of a window to a cryogenic (≈80 K) liquid CO (LCO) chamber (total volume ≈1 cm$^3$). The rapid laser-induced formation of a plasma at the window-ablator interface drives a picosecond-rise shock[16] through the ablation layer and launches a shock-wave into the LCO sample. As a result, the pressure increases rapidly in a cylindrical sample region of ≈50 μm diameter and length ≈5 μm (volume ≈$10^4$ μm$^3$) corresponding to the laser beam spot size and propagation distance of the shock-wave over the duration of the laser pump, respectively (see also "Methods" section and Supplementary Note 1).

Supplementary Fig. 2 shows several points along the measured shock Hugoniot of LCO for pressures below approximately 16 GPa, corresponding to a maximum laser pump energy of ≈220 μJ; the data are in good agreement with the gas-gun shock-wave measurements of Nellis et al.[27]. For shots at higher (≈300 μJ) pump energy the sample reflectivity falls to zero within experimental error over hundreds of picoseconds— see Fig. 2. The loss of signal precluded a direct shock pressure estimate, and it is likely indicative of liquid carbon nanodroplet formation (see below). Conventional scaling of shock pressure as a function of intensity[28] puts the pressure for the 300 μJ shots at around 20 GPa, with an estimated shock temperature of ≈5300 K[11,27,29], i.e. the state is in the liquid region of the carbon phase diagram[10].

A plot comparing the normalized intensity for an average of data over 300 μJ (>16 GPa) shots and a corresponding average over 220 μJ (10–16 GPa) shots is shown in Fig. 2. Previous work at comparable and higher pressures[13] in a (carbon-free) hydrogen peroxide system did not exhibit the strong loss of reflectivity observed in the 300 μJ shots, nor is this behavior exhibited in aluminum under similar conditions[16]. Although the reflectivity of Al does vary with pressure, the reflectivity change we observe for shots below 16 GPa is consistent with previous observations at pressures up to ≈30 GPa[16,30,31]. Thus, additional decay in the return signal for pressures above 16 GPa cannot be accounted for by loss from the ablator and must be due to attenuation in the compressed CO sample itself. Rapid chemistry behind the shock front of organic explosives is believed to generate a thin (nm scale) semi-metallic layer[32], but its possible occurrence has not played a discernable role in previous ultrafast experiments[13,25]. We, therefore, conclude that the signal disappearance at high shock pressures is due to processes occurring in the bulk of the compressed CO sample, specifically the precipitation of liquid carbon nanodroplets. Their estimated volume fraction from thermochemical calculations and MD simulations (see below) is ≈10%, and therefore they likely have a large effect on the optical properties of the compressed material. This conclusion is supported by a comparison of the time-dependent experimentally measured reflectance at >16 GPa pressures with direct estimates

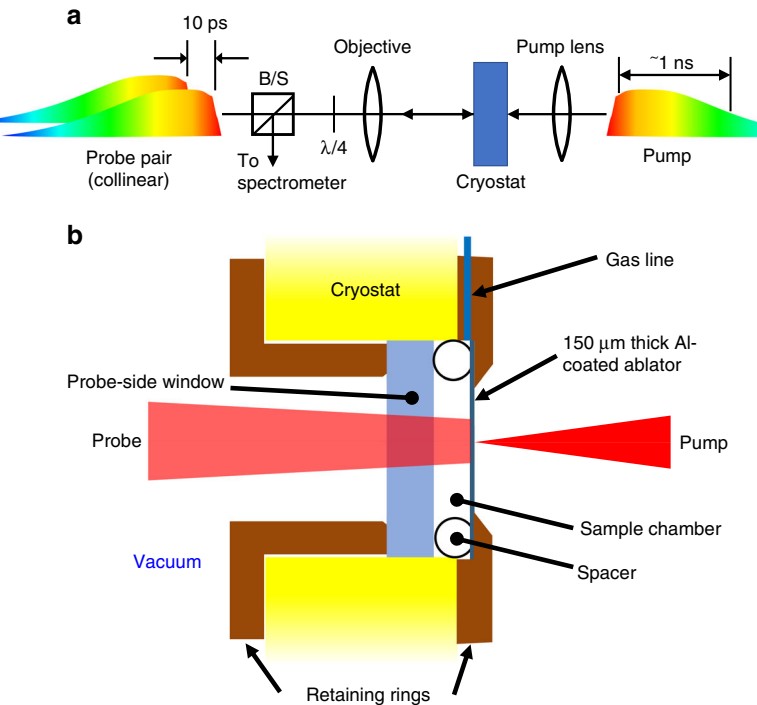

**Fig. 1 Schematic of the experimental setup. a** A focused ≈1 ns duration pump impinges from the right, while a pair of similar duration low energy pulses from the left characterize the shocked state. The reflected probes are phase-shifted by a shock wave in the sample, and the phase shift is measured via spectral interferometry, providing wave speeds. B/S is a polarizing beam splitter, and λ/4 is a quarter waveplate set to rotate the polarization by 90 degrees in double pass. **b** Close-up of the sample, where the focused (≈50 μm diameter) laser pulse drives a shock-wave into the Al ablator. The shock-wave propagates through the ablator and then compresses a small region (≈$10^4$ μm³) of the cryogenic LCO sample; the volume of the whole sample is ≈1 cm³.

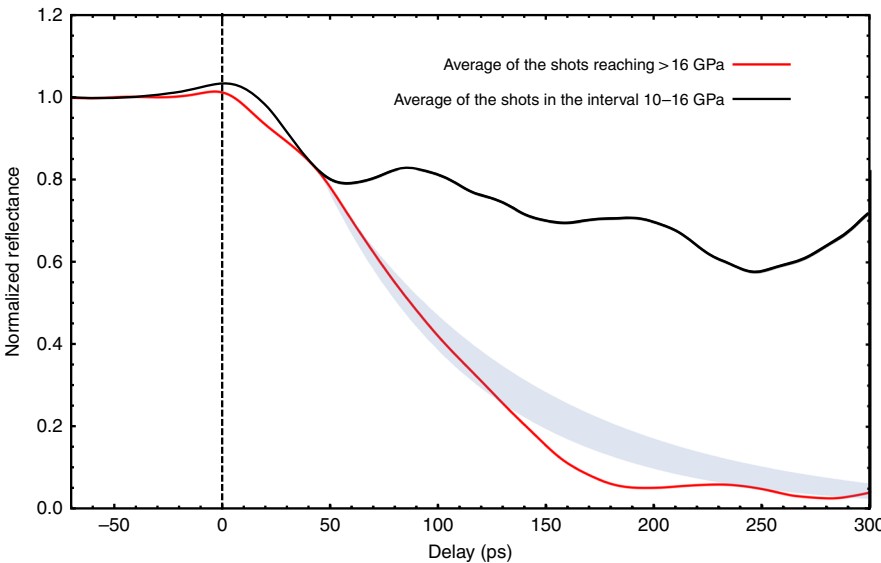

**Fig. 2 The normalized reflectivity of the sample in the pump region as a function of time for shots above and below 16 GPa.** The vertical dashed line shows the shock-wave arrival time at the ablator/LCO interface. The light gray band shows the calculated reflectance assuming a uniform volume fraction of carbon nanoparticles, where the absorption increases with the thickness increase of shocked material via shock propagation; the bandwidth is due to uncertainties in the dielectric function of liquid carbon, and accordingly in the estimate of the absorption depth. We assume loss due to the formation of carbon nanoparticles begins around 50 ps, where the reflectance of the >16 GPa data (5 shot average) significantly deviates from the 10–16 GPa data (9 shot average)—see Supplementary Discussion for further details. Source data are provided as a Source Data file.

based on the expected absorption loss through a suspension of liquid carbon nanoparticles—see Fig. 2, the Supplementary Discussion, and Supplementary Fig. 3.

This sub-nanosecond nanocarbon formation scenario is strongly reinforced by recovery of carbon nanoparticles from the sample chamber after the experiment is complete and LCO evaporates as the cryostat is brought to room temperature. Despite the small volume of compressed material per shot and only tens of shots taken for this experiment, carbon nanoparticle agglomerates were recovered from the internal surfaces of the

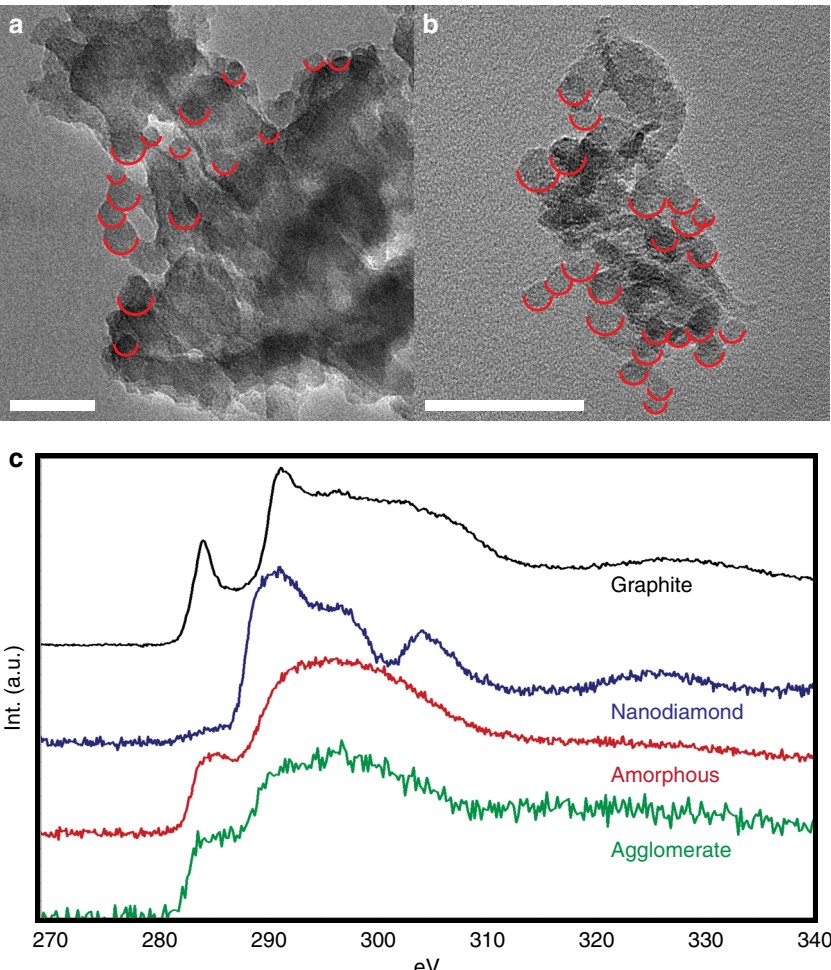

**Fig. 3 TEM characterization of recovered carbon agglomerates. a, b** Images of carbon nanoparticle agglomerates containing globular domains recovered from the inside of the sample cavity after LCO evaporation. Semicircles are placed as an aid to identify domains, while (in the top half of the semicircle) not obscuring edges. Measured domain sizes range from 5 to 30 nanometers. Supplementary Fig. 5 shows a comparison of the above images with and without markings. **c** Carbon K edge EELS spectra from the carbon agglomerate and reference carbon materials (here, amorphous corresponds to the lacey carbon of TEM grids); agglomerate spectrum matches well that of amorphous carbon. Scale bars are 50 nm. Source data are provided as a Source Data file.

sample chamber. Transmission electron microscope (TEM) images of these agglomerates are shown in Fig. 3a, b (see also Supplementary Note 2). They appear to consist of markedly globular, overlapping domains, with sizes ranging from approximately 5 to 30 nm. Carbon K edge electron energy loss (EELS) spectroscopy scans of the agglomerates lack both the strong $\pi^*$ resonance around 285 eV and structure in the $\sigma^*$ region starting around 291 eV typically observed in graphite. Likewise, the sample lacks the dip at the second gap around 302 eV associated with diamond. Indeed, the agglomerate EELS spectrum bears very close resemblance to that of an amorphous carbon reference as collected from the lacey carbon of a TEM grid. Diffraction data (see Supplementary Note 2 and Supplementary Fig. 4) also indicates that the agglomerates are amorphous carbon, as no crystalline reflections are observed. Raman spectroscopy, which was performed on the closed chamber prior to residue removal, was also consistent with the above findings as it indicates the presence of the D and G bands typical of disordered carbon[33,34] (see Supplementary Note 3 and Supplementary Fig. 6).

**Molecular dynamics simulations**. The experimental results provide direct evidence for sub-nanosecond nanocarbon synthesis via ultrafast compression but do not give information on the atomic-scale mechanisms that control it. MD simulations can supply an atomistically resolved view of this process but are challenging since relevant scales span over three orders of magnitude: from the molecular scale, where bond formation and breaking events occur, to the condensed domains scale, where formation and coarsening dynamics determine the system evolution. To overcome this, we have developed a reactive force field capable of first-principles accuracy for systems under extreme conditions[35,36] (including liquid carbon[35]), which is computationally efficient and enables multi-million atom simulations that can reach nanosecond time-scales (see "Methods" section for further details).

Here we employ *NVT* MD simulations to study CO at 6500 K and 2.5 g/cm$^3$ (p ≈ 30 GPa), well into the regime where Nellis et al. hypothesized the formation of carbon condensates[11,27]. Previous studies have shown that detonation-synthesized carbon condensates are typically of the order 5–10 nm in diameter;[3,8,10,37] at the same time, thermochemical calculations suggest that for CO at these conditions they should occupy ≈10% of the system volume[11,27,29]. To accommodate these constraints our simulation contained 1.25 million atoms in a 22.5$^3$ nm$^3$ simulation cell, i.e. large enough to observe the formation of a significant carbon nanoparticle population. We note that while the length scale of our computer simulations is by necessity much smaller than that of the ultrafast shock experiments, the time scale, of order 100 s of picoseconds, is essentially the same.

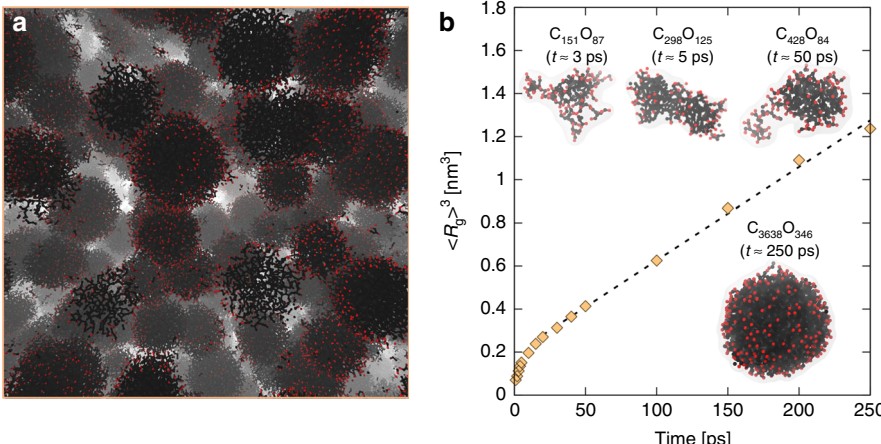

**Fig. 4 MD structure and evolution of carbon clusters. a** Snapshot of the 1.25 million atom simulation at 0.25 ns. Carbon and oxygen atoms are shown as black and red, respectively; only atoms participating in cluster formation are shown, and connections between atoms are drawn only as a guide to the eye. **b** Average cluster size evolution ($R_g$— radius of gyration) as a function of time; representative cluster snapshots are provided for various times along the simulation. A linear fit to $20 \leq t \leq 250$ ps is given by the dashed black line.

Simulations were initialized as an equilibrated fluid of CO molecules at $2.5\,g/cm^3$ and 100 K, with the temperature instantaneously increased to the target value of 6500 K. As shown in the timeline of Fig. 4, $C_{2n}O_n$ clusters of roughly 250 atoms begin to form as early as 3 ps. Consistent with previous DFT studies[38], these early clusters appear to be covalently bonded, with an extended, low density shape and oxygen distributed homogeneously throughout. They continue to grow and merge to form larger seeds, eventually densifying into small liquid carbon nanodroplets by expelling interior oxygen. For the duration of the simulation, the larger clusters continue to grow through Ostwald ripening, i.e. at the expense of the smaller clusters which shrink through dissolution and disappear. Consistent with this observation we find that after an initial transient regime cluster growth follows $t^{1/3}$ behavior, corresponding to coarsening driven by the evaporation-condensation mechanism[39]—see Fig. 4. In summary, the nanocarbon synthesis sequence that emerges from the simulations appears to be oxygen-rich polymeric seeds formation, followed by densification and oxygen elimination, and particle growth via evaporation-condensation. At late times the nanodroplets are pure molten carbon[35,40] with an oxygen-decorated surface, occupying ≈10% of the volume (a typical cluster is described in Supplementary Note 4 and shown in Supplementary Fig. 7). We expect them to exhibit metallic character at these conditions and thus be responsible for the loss of probe signal observed in the experiments. It is likely that, in the experiments, the end of the laser-pump leads to arrest of the cluster growth due to rapid decompression, cooling and mixing in the sea of cryogenic LCO. The evolution of the system following this quench may have a significant effect on the properties of the recovered nanoparticles, including size, structure, surface oxygen enrichment, etc. We believe in fact that the high cooling rates likely achieved, $\geq 10^{12}$ K/s (i.e., thousands K cooling over ≈1 ns of decompression time) may inhibit crystallization of the liquid carbon nanodroplets, which is consistent with the amorphous character of the carbon nanoparticles recovered.

## Discussion

Our combined experimental and simulation results show that high pressure shock synthesis of carbon nanoallotropes need not be limited to detonation, nor is it necessarily a slow process. The MD simulations reveal the sub-nanosecond, atomic-scale processes governing carbon nanoparticle formation and evolution and identify classical evaporation-condensation as the dominant mechanism for nanoparticle growth on these time-scales; we expect that the evolution sequence that emerges from the simulations, likely followed at later times by Brownian aggregation[41], is broadly relevant for nanocarbon condensation in a high pressure–high temperature chemically reactive system. By decoupling shock generation from chemical properties of the precursor, the present ultrafast experimental technique may open the door to tailor-made carbon nanoparticles through better control over particle size, structure, and composition. The method has several advantages over previous high pressure synthesis approaches—including high repetition rate and enhanced process control through the laser-pump profile, precursor composition and initial state, and it could be directly guided in the future by MD simulations. Moreover, the use of a liquid precursor enables high-throughput flow synthesis, which may be a significant advantage for both research and industrial applications.

## Methods

**Ultrafast velocimetry**. Picosecond resolution velocimetry in these experiments is obtained using a 1 ns duration, ~22 nm FWHM bandwidth chirped pulse from a conventional 800 nm Ti:sapphire chirped pulse amplifier system employing a high-dispersion custom-built stretcher followed by a regenerative amplifier, but with no compressor. A knife-edge is placed at the symmetry plane of the stretcher to create a sharp rise in the intensity at the leading edge of the chirped pulse, facilitating a fast rise in the ablation-generated shock wave. The ≈800 µJ amplified pulse is split into two beams: a strong pump (with hundreds of microjoule energy) and a weak probe (with microjoule energy). The probe is further split via a white-light interferometer into two identical, 10 ps delayed colinear pulses as shown in Fig. 1. These pulses illuminate the ablator on the CO side over an approximately 200 µm diameter spot size. The pump is focused to ≈50 µm diameter onto the (opposite) glass side of the ablator in the center of the probe region. The probe region is imaged using a long working distance Mitutoyo 10× microscope objective onto the slit of a Princeton Instruments 300 mm imaging spectrometer coupled to a Pixis 100 camera, with the pump region centered on the slit. Time resolution is obtained via the wavelength-time correlation in the chirped pulse. The resulting spatially resolved spectral interferogram encodes phase shift information between the two 10 ps delayed probe pulses, analogous to commonly employed velocimetry techniques such as VISAR[42], and this signal measures the piston and shock velocity of the pump-generated shock via beating between the Doppler-shifted probe reflections from the piston (Al/CO interface) and the shock front, and a differential phase offset from the baseline (see Armstrong et al.[13,25,26]. and Supplementary Note 1 for further discussion).

**Time-resolved reflectometry**. Reflectivity data are obtained by taking the ratio between the probe intensity at the center of the pump region and outside the pump region for the same shot, to normalize time variation in the probe intensity. Raw experimental intensity data vary sinusoidally (see ref. [43].) and are smoothed (prior

to taking the ratio) by Fourier transforming the intensity distribution, shifting the carrier to DC, and inverse Fourier transforming to recover the time domain intensity without a sinusoidal carrier. Comparisons between the raw data and the smoothed data show excellent correspondence between the smoothed data and the intensity envelope of the raw data.

**Cryostat**. The cryostat is a Janis ST-500-UC Microscopy Cryostat with optical access to both sides of the sample holder. The sample cell comprises the as-built sample holder of the cryostat supporting custom-built copper retaining rings which mount a 3 mm thick fused silica window on the probe side and a 150 μm thick cover slip window on the pump side. A ~1 μm thick Al ablator coats the internal surface of the pump-side window. The sample thickness is ~2 mm with the windows separated and pushed against the retaining rings by a Teflon ring with a break to allow gas to enter the center of the cell. The pump-side retaining ring connects to a gas inlet and outlet via a feed-through on the cryostat to flow gaseous CO through the cell, which is condensed to liquid at approximately 80 K for the experiment using liquid nitrogen as a coolant. CO is toxic—waste CO is evacuated via a custom-built exhaust system. The windows are sealed to the retaining rings and the retaining rings are sealed to the cryostat sample holder with indium wire gaskets and the rings are compressed with aluminum screws through the entire fixture.

**Electron microscopy**. TEM samples were collected through mechanical contact of $SiO_2$ membranes (Ted Pella, Inc.) with the window surfaces of the experimental cell. A clean $SiO_2$ membrane was also characterized in the TEM as a control to verify that material observed on the collected sample grid originated from the sample chamber. TEM analysis was conducted using a JEM-2100F (JEOL Ltd.) analytical electron microscope operating at 200 kV, equipped with an Orius SC1000 CCD (Gatan, Inc.) and a GIF Tridiem (Gatan, Inc.). Images and diffraction patterns were collected with both detectors, and electron energy loss spectroscopy (EELS) data were collected with the GIF using dispersions of 0.1 or 0.2 eV per channel. EELS spectra were collected with the microscope operating in STEM mode using a beam spot size of 0.7 nm and convergence and collection angles of 12 mrad. All TEM data was acquired and analyzed using DigitalMicrograph (Gatan, Inc.). EELS spectra were processed in DigitalMicrograph for background removal and alignment.

**Raman spectroscopy**. Subsequent to shock compression and return of the cell to room temperature, but prior to opening the sample cell, Raman studies were performed using the 514.5 nm line of an Ar laser in the backscattering geometry. The laser probing spot dimension was about 2 μm. Raman spectra were analyzed with a spectral resolution of 2 cm$^{-1}$ using a single-stage grating spectrograph equipped with a CCD array detector. Ultra-low frequency solid-state notch filters allowed us to measure the Raman spectra down to 10 cm$^{-1}$ [44]. The laser intensity was kept <2 mW to minimize sample heating.

**Molecular dynamics simulations**. The recently developed Chebyshev Interaction Model for Efficient Simulation (ChIMES)[35,36] was employed for simulations in the present work. ChIMES is a generalized many-body classical reactive force-field constructed from linear combinations of Chebyshev polynomials, machine learned to Density Functional Theory (DFT)[45] molecular dynamics simulations. The substantial flexibility afforded through the use of a polynomial basis enables ChIMES to serve as a high-fidelity proxy for DFT while providing a four-orders-of-magnitude increase in simulation efficiency and linear system size scalability[35,36]. The ChIMES CO model used herein contains explicit two- and three-body inter-actions and has been shown to yield good agreement with the DFT-calculated structure, dynamics, and energetics at the conditions studied in this work[35]. As a succinct example of its chemistry accuracy (at 6500 K and 30 GPa (i.e. the state point examined in this work) for a system of 64 CO molecules at $t = 6$ ps DFT predicts a total of $0.67_2$, $1.20_3$, and $0.006_5$ C–C, C–O, and O–O bonds per initial molecule, respectively, whereas ChIMES predicts $0.613_8$, $1.23_1$, and $0.006_1$. Simulations were conducted using a version of LAMMPS[46] modified to use the ChIMES potential. They were run in the $NVT$ (canonical) ensemble with a Nose-Hoover thermostat[47] and a 0.2 fs timestep and were initialized as an equilibrated fluid of CO molecules at 2.5 g/cm$^3$ and 100 K; the temperature was then instantaneously increased to the target value of 6500 K, resulting in a pressure of ≈30 GPa. Initial velocities were assigned according to a random Gaussian distribution yielding the target system temperature. Clusters were defined as any collection of greater than 10 carbon atoms where each atom was within 1.9 Å of at least one other cluster member; their radius of gyration was used as a measure of their size.

## Data availability
The experimental datasets generated and/or analyzed in the current study are available from MRA on reasonable request.

## Code availability
MD simulations were performed using the LAMMPS software package, which is available at https://lammps.sandia.gov. The force-field model employed in this work is available at https://chemrxiv.org/s/e482138dc3fccfa55ee9.

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

## Acknowledgements

This work was performed under the auspices of the US Department of Energy by Lawrence Livermore National Laboratory under Contract DE-AC52-07NA27344 and funded by LLNL Laboratory Directed Research and Development Program project 17-ERD-011 with S.B. as principal investigator. We thank M.V.P. Altoe and S. Aloni for the use of and assistance with the TEM. We also thank R. Swan for CAD design and fabrication of the retaining rings, and D. Qu, P. Grivickas, and J. Jeffries for help assembling and operating the cryostat, T. Li for a preliminary TEM analysis and W. Kuo for useful discussions on MD modeling. Work at the Molecular Foundry was supported by the Office of Science, Office of Basic Energy Sciences, of the U.S. Department of Energy under Contract No. DE-AC02-05CH11231.

## Author contributions

S.B. conceived and directed the study. M.R.A. designed and built the experimental set-up, performed the ultrafast experiments and data analysis, R.K.L. developed the force-field and carried out the MD simulations, N.G. contributed to the force-field development and executed DFTB simulations, M.H.N. collected and analyzed TEM data, E.S. performed Raman measurements, L.E.F. contributed to the force-field development, J.M.Z. contributed to the experimental design, S.B. performed theoretical calculations and contributed to the analysis and interpretation. All authors participated in scientific discussions.

## Competing interests

The authors declare no competing interests.
