## [Peer Review File · Nature Communications]

REVIEWERS' COMMENTS:

Reviewer #2 (Remarks to the Author):

The authors have made substantial improvements to the manuscript. Thus, I believe the manuscript is now worthy and ready for publication in *Nature Communications*.

Good work!

Reviewer #3 (Remarks to the Author):

The authors have written a detailed rebuttal letter that, in my opinion, provides an appropriate and well-argued response to all items. Another referee and I had raised a similar question (concerning the "tunability" of the reaction conditions and the products), but I agree with the authors that such studies will require very much additional work and that there will be value in communicating the present results to the community as they are. It is also understood that this is (as the authors call it) a "conceptual" study that has value and provides insight independent from possible application of their materials.

The authors have made satisfactory changes to the manuscript during revision and thus I strongly support publication of this interesting work in its present form.

REVIEWERS' COMMENTS:

Reviewer #2 (Remarks to the Author):

The authors have made substantial improvements to the manuscript. Thus, I believe the manuscript is now worthy and ready for publication in Nature Communications.

Good work!

Reviewer #3 (Remarks to the Author):

The authors have written a detailed rebuttal letter that, in my opinion, provides an appropriate and well-argued response to all items. Another referee and I had raised a similar question (concerning the "tunability" of the reaction conditions and the products), but I agree with the authors that such studies will require very much additional work and that there will be value in communicating the present results to the community as they are. It is also understood that this is (as the authors call it) a "conceptual" study that has value and provides insight independent from possible application of their materials.

The authors have made satisfactory changes to the manuscript during revision and thus I strongly support publication of this interesting work in its present form.

Thanks to both reviewers for positive remarks on our work, and for their thoughtful, comprehensive reviews.